# Anaxyelidae of Karatau: 100 Years After [note 1]

**DOI:** 10.3390/insects16090943

**Published:** 2025-09-09

**Authors:** Dmitry S. Kopylov, Alexandr P. Rasnitsyn

**Affiliations:** 1Institute of Zoology, 93 Al-Farabi Avenue, Almaty 050060, Kazakhstan; 2Paleontological Institute, Russian Academy of Sciences, 123 Profsoyuznaya Street, Moscow 117647, Russia; alex.rasnitsyn@gmail.com; 3Natural History Museum, London SW7 5BD, UK

**Keywords:** Hymenoptera, Syntexinae, woodwasp, wasp, insect, Kazakhstan, Karabastau, Galkino, Mikhailovka, Jurassic, fossil, new taxa

## Abstract

The family Anaxyelidae is a relict group of woodwasps, represented in the extant fauna by a single surviving species. In stark contrast to its dwindling present-day diversity, Anaxyelidae was among the most prosperous and diverse hymenopteran groups in the Late Mesozoic. Taxonomically, the family is divided into two subfamilies: Anaxyelinae, dominant in the Jurassic, and Syntexinae, which prevailed in the Cretaceous and survive to the present. Until now, only one Jurassic record of Syntexinae was known—from the Middle Jurassic of China. During our 2025 fieldwork, we discovered the second Jurassic representative of Syntexinae in the Late Jurassic deposits of Karatau (Kazakhstan). The fossil sites of the “Jurassic Lake of Karatau” complex have seen very limited excavation over the past 50 years. During our expedition, we visited all the key sections of Karatau. These outcrops are in satisfactory condition and merit further exploration.


*This work is dedicated to the memory of the eminent Russian entomologist Andrey Vasilyevich Martynov (1879–1938), the founder of the Russian school of paleoentomology and the first researcher of fossil insects from Karatau.*


## 1. Introduction

Today, the “Jurassic Lake of Karatau” ranks among the richest Lagerstätten for fossil insects of the Jurassic period. With over 800 recorded species [1], it rivals only Daohugou (Middle Jurassic of China) in terms of described insect diversity, leaving all other Jurassic assemblages far behind.

The fossil flora of Karatau Jurassic deposits had already attracted attention by the late 19th and early 20th centuries [2,3]. Yet the works of that era leave us guessing which outcrops were actually studied and how they relate to today’s understanding of the Karabastau Formation. The fossil insect fauna itself was brought to light much later, in 1921, when mining engineer A.A. Aniskovich made the first discoveries near the village of Galkino [4]. The finds caused a sensation in the Soviet paleontological community. Throughout the 1920s, dozens of researchers from various institutions descended on the Jurassic beds of Karatau. In 1923, fossil remains were discovered in the Karabastau and Chugurchak ravines, and in 1925, the now-famous Mikhailovka site was found (Toponyms such as Mikhailovka, Galkino, and Karabastau are common across the territory of the former USSR and in some cases even refer to other fossil localities; in this work, these names are used exclusively to designate sections within the Karabastau Formation). It quickly became the most productive locality of the Karabastau Formation and yielded the vast majority of the region’s fossils.

The first formal paleoentomological excavations in Karatau were led by the outstanding Russian scientist Andrey Vasilyevich Martynov. Based on his expedition material, he published a landmark series titled “To the knowledge of fossil insects from Jurassic beds in Turkestan”—a seven-part opus describing over 50 insect species [5,6,7,8,9,10]. At the time, the Mikhailovka locality had not yet been discovered, so nearly all the material came from Galkino. It was in this series, exactly 100 years ago, that the first fossil anaxyelid—*Anaxyela gracilis*—was described, along with the family Anaxyelidae itself. Soon after, Cockerell [11] added *Anaxyela martynovi* from the same deposits, although this species was later synonymized [12].

The real turning point in the study of Karatau fossil fauna came in the 1960s. During that decade, numerous expeditions from the Paleontological Institute of the USSR Academy of Sciences collected tens of thousands of fossil insect specimens. It was during this golden era that A.P. Rasnitsyn published his series of works, describing the rich diversity of Karatau anaxyelid fauna—around 20 species [12,13,14]. He also demonstrated a close relationship between the Jurassic anaxyelids and the extant, monotypic family Syntexidae Benson, 1935, which was subsequently merged into Anaxyelidae [12].

After the surge of taxonomic discoveries in the 1960s, research on the Jurassic deposits of Karatau slowed, and the known diversity of anaxyelids remained largely static. Over the next five decades, only five papers were published, describing a mere seven new Mesozoic species. Nevertheless, these contributions expanded the family’s known range and provided the first fossil evidence for the extant subfamily Syntexinae [15,16,17,18,19].

A true renaissance in anaxyelid research began in 2018 and is still ongoing. The momentum has been driven largely by discoveries in major Mesozoic insect deposits such as Burmese amber, Daohugou, and Khasurty, though it has not stopped there. In just seven years, more descriptive works on fossil anaxyelids have been published than in the entire previous century—and the documented diversity of the family has doubled [20,21,22,23,24,25,26,27,28,29,30]. We also know for certain of at least four additional papers by various authors that are currently in press. This new wave of research also reached the Karatau fauna: several previously overlooked species were rediscovered and described from the collections of the Paleontological Institute of the Russian Academy of Sciences.

Following the dissolution of the Soviet Union, the flow of new material from Karatau almost ceased. The lack of legal mechanisms for exporting fossils has deterred international researchers, while the local paleoentomological school remains virtually nonexistent.

In 2025, for the first time in many years, a dedicated paleoentomological expedition was organized to the Karatau section—led by the Institute of Zoology of the Republic of Kazakhstan with strong support from the Paleontological Institute of the Russian Academy of Sciences. The main goal was to survey the condition of key fossil localities in the Western Tien Shan. As a result, the collected material was modest in quantity but covered all the key sites of the region. Remarkably, all four main localities of the “Jurassic Lake of Karatau” were revisited.

Among several hundred collected specimens, only one anaxyelid was found, at Galkino. The specimen is not well preserved, but it clearly belongs to the subfamily Syntexinae, which had not previously been recorded from Karatau. Until now, the Jurassic record of Syntexinae included only a single species: *Daosyntexis primus* Kopylov et al., 2020, from Daohugou [22].

In this paper, we describe a new genus and species (and a new subfamily record for Karatau) and offer a brief assessment of the current condition of the key localities within the “Jurassic Lake of Karatau” fossil complex.

## 2. Materials and Methods

This study is based on a single specimen discovered at the Karatau-Galkino locality. The fossil was collected during a 2025 field expedition of the Institute of Zoology of the Republic of Kazakhstan and is housed in its collection in Almaty under the catalog number IZRK 1004/213A.

Previously described species discussed in the Discussion section and provided in Appendix A originate from Karatau-Galkino and Karatau-Mikhailovka. The type material of these species is deposited in the collection of the Paleontological Institute of the Russian Academy of Sciences (PIN RAS), Moscow.

A detailed characterization of the localities is provided in the Discussion. Appendix A includes photographs of all type specimens of Anaxyelidae from Karatau, held in both collections, and available online at https://doi.org/10.5281/zenodo.16977336.

The specimen was collected using standard extraction methods for compression fossils: shale blocks were split along bedding planes using targeted blows with the pointed edge of a 300-g hammer. Fossils were searched for on the split surfaces with a 10× jeweler’s loupe. To prepare the fossils for transport, the surrounding matrix was trimmed with pliers or a battery-powered angle grinder, then each sample was wrapped in soft toilet paper or paper towels and securely packed in plastic containers.

Initial laboratory preparation was performed with a Dremel 290 engraver. Fine cleaning was done manually using preparation needles handmade from sharpened files and entomological pins.

The usual contrast-enhancement method for compression fossils—wetting the surface with alcohol—proved ineffective for this specimen due to the near-complete absence of pigmentation. In this case, oblique lighting was found to be more effective. Photographs of the specimen were taken with an ICOE SZ1800MT (Ningbo Icoe Commodity Co., Ltd., Ningbo, China) stereomicroscope equipped with a PlanApo 1× objective and an ICOE HX4K-16A camera (Ningbo Icoe Commodity Co., Ltd., Ningbo, China), using the ImageView x64 v.4.12 software. Photographs of the PIN RAS collections shown in the appendix were taken with a Leica M165C stereomicroscope (Leica Microsystems, Wetzlar, Germany) and Leica DFC-425 camera (Leica Microsystems, Wetzlar, Germany). Line drawings were prepared in Inkscape v.1.4.

Wing venation terminology follows Rasnitsyn [12], with the addition of numbered vein abscissae (Figure 1D).

This work and the included nomenclatural acts are registered in ZooBank (https://www.zoobank.org, accessed on 2 September 2025) under the LSID: urn:lsid:zoobank.org:pub:D9BBA3C6-369B-49AD-B76F-F0A1C8681753.

## 3. Results


**Systematic palaeontology**



**Order Hymenoptera Linnaeus, 1758**



**Superfamily Siricoidea Billberg, 1820**



**Family Anaxyelidae Martynov, 1925**



**Subfamily Syntexinae Benson, 1935**



**Genus *Karasyntexis* Kopylov et Rasnitsyn, gen. nov.**


LSID: urn:lsid:zoobank.org:act:CC3F201E-5221-4F46-9F0E-1DFE4FE43882

**Etymology.** The genus name is derived from the type locality, Karatau, and the genus *Syntexis*. Gender feminine.

**Type species**. *Karasyntexis martynovi* sp. nov.

**Species included**: Type species only.

**Diagnosis**: Forewing with Sc absent; pterostigma almost completely sclerotized, with a small desclerotized spot in apical half; 1r-rs and 2r-rs fully developed, tubular; 2r-rs joins pterostigma near its apical 2/5; cell 2r 1.5× as long as wide, as wide basally as apically; cell 3r 1.2× as long as 1r and 2r together; 1-M 0.7× as long as 1-Cu; Rs+M bifurcating before 1m-cu; 5-M 1.1× as long as 3r-m. Hind wing with cell r closed. The ovipositor is 0.5× as long as the forewing, and its apical section is 0.6× as long as the basal one.

**Comparison**. With both r-rs completely developed, Rs+M bifurcating well before 1m-cu, and 1-M shorter than 1-Cu, the new genus is the most similar to *Daosyntexis* and *Hemisyntexis*. It differs from *Daosyntexis* in the absence of Sc; the much longer 5-M (1.1× as long as 3r-m in *Karasyntexis* vs. 0.14× in *Daosyntexis*); 2r-rs joins the pterostigma near its apical 2/5 (vs. near its basal 2/5 in *Daosyntexis*); and the ovipositor apical section is 0.6× as long as the basal one (vs. 0.4× in *Daosyntexis*). It differs from *Hemisyntexis* in having the pterostigma desclerotized apically (vs. widely desclerotized basally in *Hemisyntexis*) and the ovipositor 0.5× as long as the forewing (vs. 1× as long as the forewing in *Hemisyntexis*).

***Karasyntexis martynovi* Kopylov et Rasnitsyn, sp. nov.** (Figure 1).

LSID: urn:lsid:zoobank.org:act:BE0DC4C8-C28A-415A-943F-6A1BD0027B99

**Etymology.** The new species is named in memory of the Russian paleoentomologist A.V. Martynov, who was the first to describe fossil anaxyelids.

**Material**: holotype IZRK 1004/213A, part and counterpart. Imperfectly preserved impression of a female imago in limestone shale. Body structures blurred; forewings almost complete, hindwings preserved partly, legs not preserved, antennae with segments unrecognizable, and ovipositor completely preserved. Coloration is not preserved except partially for the forewing. Karatau-Galkino, Upper Jurassic, Karabastau Fm., Zhambyl Region, Kazakhstan.

**Diagnosis**: As for the genus (vide supra).

**Description**. Mesoscutellum with pointed front edge, narrow, 1.6× as long as wide. Forewing with costal area as wide as R vein before bifurcation; Sc absent; pterostigma 4.3× as long as wide, 0.6× as wide as cell 2r width, with 2r-rs near its apical 2/5; 1-Rs 2.6× as long as 1-M; Rs+M 3.7× as long as 2-M; 1r-rs 1.1× as long as 2r-rs, 1r-rs parallel to 2r-rs; cell 2r 1.5× as long as wide, as wide basally as apically; 3r-m slightly inclined; 5-M 1.1× as long as 3r-m; 1-Cu 0.5× as long as 2-Cu; cell 1mcu 2.0× as long as wide; 1m-cu straight; 2-1A 0.3× as long as 3-1A. Hindwing with cell r 4× as long as wide. The ovipositor is 0.5× as long as the forewing, and its apical section is 0.6× as long as the basal one.

Body length 7.7 mm: head—1.6 mm, thorax—ca. 2.2 mm, abdomen—ca. 3.9 mm, ovipositor length—2.8 mm: basal section—1.78 mm and apical section—1.04 mm. Forewing length: preserved—5.0 mm, estimated—5.5 mm; width—2.1 mm.

## 4. Discussion

### 4.1. Current State of Key Sections of the “Jurassic Lake of Karatau” Lagerstätte

The Jurassic deposits of the Karabastau Formation stretch in a narrow band nearly 200 km long along the northern slope of the Karatau Range. According to recent data, these deposits date to the Kimmeridgian, although a slightly older, Callovian age cannot be ruled out [31]. R.F. Gekker [4] identified four key fossiliferous outcrops in Karatau: Galkino, Mikhailovka, Karabastau, and Chugurchak. These localities are spread over a distance of about 50 km and likely represent not only different geographic parts of Lake Karatau but also different temporal phases of its existence (Figure 2). Unfortunately, the early descriptions of the Karatau sections do not include excavation coordinates; however, they provide sketch maps and geological columns sufficient to locate the outcrops mentioned in the literature. During our fieldwork in 2025, we succeeded in locating and visiting all the sites described by Gekker.

**Galkino** (N42.69 E70.41, Figure 3A) is situated in a valley about 3 km west of the village of Konyrtobe (formerly Uspenovka or Galkino). It was the first locality discovered among those belonging to the Karabastau Formation and is geographically isolated from the main cluster of Karabastau sites. The main challenge of working at Galkino is the absence of large outcrops with insect-bearing shales. Instead, scattered blocks of shale are occasionally found along small streams crossing the valley. During our fieldwork, we examined four separate points in Galkino, some of which have been fully exhausted. Fossil preservation at Galkino is generally good. Fossils are found both in extremely thinly laminated “paper” shale (less than 1 mm) and in more massive shales. In our experience, both preservation quality and fossil density tend to be better in the latter. The land around the site is under private ownership (long-term lease by local farmers), but we encountered no difficulty in obtaining permission to work there. The main challenge is the patchy distribution of fossiliferous zones, which necessitates ongoing prospection and hinders correlation between individual sites.

**Mikhailovka (Aulie)** (N42.90 E70.00, Figure 3B) is the central and most significant locality of the Karabastau Formation, yielding the most important finds and the bulk of taxonomic material. It lies in the hills about 3.5 km southwest of the village of Aktas. The village of Koshkarata (formerly Mikhailovka until 1992), which gave the site its name, is in fact located over 7 km from the fossil locality. Insect-bearing shales are exposed at several accessible points that are easy to locate in the field. Previous reports have raised concerns about the extensive destruction or even complete loss of the Mikhailovka outcrops due to illegal fossil collecting [3]. However, in our assessment, the site still contains enough fossil-rich sediment to support many more full-scale expeditions. The most productive beds may no longer be as accessible as in the early 20th century, and overburden removal may be required in some areas, but the site’s overall potential remains high. Formerly protected as part of the Aksu-Zhabagly Nature Reserve, the Mikhailovka site has recently been reassigned to the Syrdarya-Turkestan Regional Nature Park due to cadastral changes. This has complicated legal access: obtaining permits for work in protected areas in Kazakhstan involves a prolonged and difficult process, even for state institutions. Since the site lies along the border between the two reserves, approval from both is now required to conduct excavations.

**Figure 3 insects-16-00943-f003:**
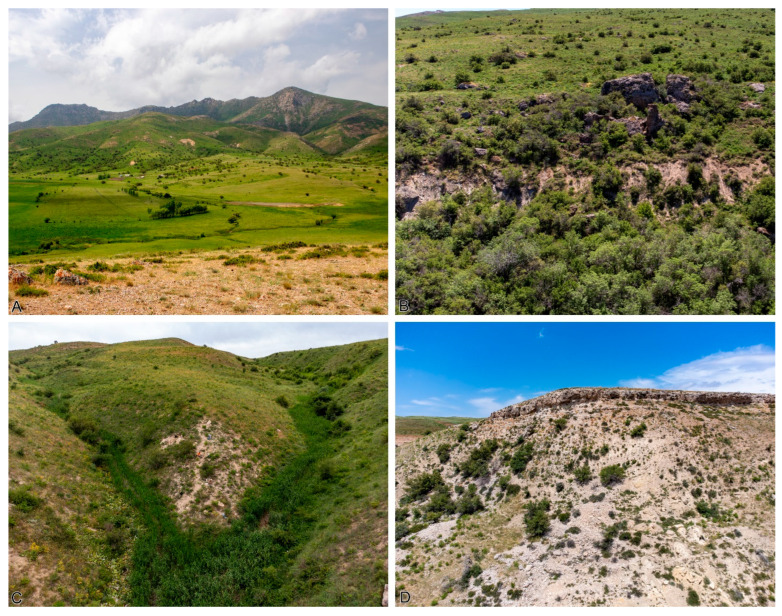
Main outcrops of the “Jurassic Lake of Karatau” as of 2025. (**A**) Galkino (numerous exposures along stream banks in the valley). (**B**) Mikhailovka. (**C**) Karabastau. (**D**) Chugurchak.

**Karabastau** (N42.95 E69.92, Figure 3C) is located in the hills 4 km south of the village of Baidibek-Ata (formerly Kitaevka), on the left slope of the Karabastau ravine. This ravine reaches depths of up to 400 m and flows into the canyon of the Koshkarata River. Jurassic beds are found only in the uppermost portion of the ravine. Gekker [4] provided a fairly detailed description of the Karabastau section, including recognizable sketches of the landscape. Nevertheless, we were unable to find insect-bearing shales in the layers described by Gekker. We located insect fossils approximately 300 m from the point he recorded, and our site lies significantly lower in the stratigraphic section. As such, we cannot be certain that our specimens originate from the exact same stratigraphic level as the classical Karabastau locality. Still, the collected material is abundant and well preserved, and we plan to study it in future publications. The Karabastau locality lies within the boundaries of the Aksu-Zhabagly Nature Reserve.

**Chugurchak** (N42.95 E69.94, Figure 3D) lies in the next ravine to the east, approximately 2.3 km from Karabastau. Here too, we were unable to find fossils in the beds described by Gekker. A small but peculiar insect assemblage was recovered from piles of rock debris left by landslides. The lithology of the insect-bearing deposits in Chugurchak is very unusual for Karatau. Fossils occur in siliceous, poorly fissile rocks. Similar layers are exposed at the base of the cliffs above the talus, but we found no fossils in situ. It appears that these rocks must undergo natural weathering before they can be split effectively. The Chugurchak site likely lies within the Aksu-Zhabagly Reserve, although the boundary maps are imprecise, and the site may also fall under the Syrdarya-Turkestan Park.

The bulk of fossil material from the Jurassic of Karatau was collected during Soviet times and is now housed in the collections of the Paleontological Institute, Russian Academy of Sciences, Moscow. The PIN collection includes about 20,000 insect fossils. By contrast, Kazakhstan’s own holdings are limited to a few small museum collections. During our 2025 expedition, we collected around 800 fossil specimens, including insects, fish, and plants. These specimens will be curated in the collections of the Institute of Zoology of the Republic of Kazakhstan in Almaty. We hope this new material will serve as a foundation for developing a national Jurassic collection in Kazakhstan.

### 4.2. A Brief History of the Anaxyelidae

Anaxyelidae is a relict family of siricoid sawflies. Its only extant representative, *Syntexis libocedrii*, is found in North America [32,33]. The first fossil anaxyelid, *Anaxyela gracilis*, was described a decade later from the Karatau-Galkino locality [7]. Initially, fossil and extant anaxyelids were regarded as separate families, which were only synonymized much later, following the accumulation of substantial paleontological material [12].

Rasnitsyn originally divided the family into four subfamilies: Anaxyelinae, Syntexinae, Kempendajinae, and Dolichostigmatinae. However, a phylogenetic analysis by Gao et al. [25] demonstrated that both Kempendajinae and Dolichostigmatinae are subordinate to Anaxyelinae, leading to their synonymization under the latter. While we are not proponents of a strictly cladistic approach to taxonomy and see limited benefit in eliminating paraphyletic groups for their own sake, we find no substantial contradiction in Gao et al.’s system and will adopt it going forward. Moreover, we must acknowledge that the earlier designation of *Kempendaja* and *Mangus* as a separate subfamily based on the presence of a longitudinal Sc vein [21,34] was a mistake. Although *Mangus* does possess a well-developed Sc stem, its presence in *Kempendaja* is likely an optical illusion caused by the taphonomy of the wing.

As a result, the family Anaxyelidae is currently divided into two subfamilies: Anaxyelinae, which includes 35 species, and Syntexinae, with 18 Mesozoic species and one extant representative. These subfamilies exhibit a striking temporal shift in taxonomic diversity. In the Jurassic, Anaxyelinae overwhelmingly dominates: 27 species of Anaxyelinae are known, compared to just two Syntexinae. Jurassic records are restricted to Karatau and Daohugou. While the taxonomic diversity of Karatau’s anaxyelids appears to be thoroughly described, Daohugou still contains a wealth of undescribed material, mostly Anaxyelinae. In the Early Cretaceous, the two subfamilies are represented in roughly equal numbers: eight Anaxyelinae species versus nine Syntexinae. In the Late Cretaceous, only Syntexinae are known: one species from Obeshchayushchiy and six species from Burmese amber. The single extant species is also a syntexine. Despite this clear shift in the taxonomic ratio, it is noteworthy that both subfamilies were already present in Daohugou—the oldest known anaxyelid fauna.

### 4.3. Biology of Anaxyelidae

Our knowledge of anaxyelid biology is based mostly on the only extant representative of the family, *Syntexis libocedrii*. This woodwasp inhabits coniferous forests of the western coast of North America (California, Idaho, Oregon, and British Columbia). A single occurrence has also been recorded on the Iberian Peninsula near Granada, though this was likely an accidental introduction via timber shipments [35]. The larvae of *Syntexis* develop in the wood of coniferous trees belonging to the genera *Calocedrus*, *Juniperus*, and *Thuja*. A notable biological feature of the species is its apparent preference for fire-damaged trees. In laboratory experiments, females showed a strong preference for scorched trunks and ignored undamaged ones [36]. The larva develops within the wood for 1–2 years; the adult lifespan remains unknown. The female has a short ovipositor and lays eggs 1–4 mm deep into wood. It prefers medium-sized trees with a trunk diameter of 15–20 cm and avoids larger trees, being unable to penetrate thick bark. A single female can lay up to 120 eggs. Females measure 8–16 mm in length; males are smaller, 6–12 mm. As a rare species targeting already fire-damaged trees, *Syntexis libocedrii* poses no threat to forestry operations [36].

Mesozoic anaxyelids are known from compression fossils and amber inclusions. As is typical in paleontology, such material offers limited insight into the biology of the organisms. In such cases, the primary approach to reconstructing fossil lifestyles is the principle of actualism, which in this case states that the biology of extinct animals should be assumed similar to that of their closest living relatives unless there is evidence to the contrary.

Mesozoic anaxyelids possess ovipositors comparable in form to those of modern representatives and likely deposited their eggs in wood (xylophagous development is typical for the whole superfamily Siricoidea). Among fossil taxa, ovipositor length varies significantly, from short forms resembling *Syntexis* (e.g., *Brachysyntexis*, *Curiosyntexis*, *Daosyntexis*) to medium (*Anaxyela*, *Parasyntexis*, *Sphenosyntexis*, *Urosyntexis*) to very long (*Kulbastavia*). The ovipositor of the new genus *Karasyntexis* falls within the short-to-medium range. It is worth noting that a long ovipositor is essential for woodwasps to penetrate tree bark, whereas the actual depth of egg deposition into the wood is less critical. This suggests that Mesozoic anaxyelids may have diverged in terms of preferred host tree species or specialized on trees of different sizes. However, we lack direct evidence, such as fossilized bore marks in wood, of anaxyelid larval activity in the Mesozoic.

Extant anaxyelids develop exclusively on gymnosperms. In the Late Mesozoic, gymnosperms dominated terrestrial vegetation, suggesting they likely provided the main host base for anaxyelids at that time. Supporting this hypothesis is the fact that anaxyelids disappear from the fossil record in the Late Cretaceous—coinciding with the global rise of angiosperms as dominant plant groups.

There is also no direct evidence that Mesozoic anaxyelids favored pyrophytic habitats. Wildfires were not uncommon in the Late Mesozoic, as indicated by charred plant layers in several localities, but we see no correlation between these layers and the presence of anaxyelid fossils. However, the absence of such a correlation does not rule out the possibility of pyrophilous preferences in woodwasps: insects are generally rarely preserved in the same layers as charred wood due to purely taphonomic reasons.

An intriguing trend is observed in their geographic distribution. In the Jurassic, Anaxyelidae are known exclusively from Asia. Starting in the Early Cretaceous, records begin to appear in Europe. In the Late Cretaceous, fossils are found only in Magadan and Myanmar (which was an isolated island in the Tethys Ocean during the Cretaceous). The sole living representative, however, occurs in North America. It is likely that the surviving population of *Syntexis libocedrii* represents the last remnant of a once widespread anaxyelid lineage that, during the Mesozoic, occupied much of Laurasia. However, there is no known fossil record of the family from the Cenozoic, and the migration routes of anaxyelids over the past 100 million years remain obscured by the incompleteness of the fossil record.

### 4.4. Anaxyelidae of Karatau

To date, 22 species of Anaxyelidae have been described from Karatau, making it the richest anaxyelid assemblage in the fossil record. Of these, 21 species belong to Anaxyelinae and one to Syntexinae. A full taxonomic list of Karatau anaxyelids is provided in Table 1.

Most of the taxonomic diversity from Karatau was described during the 1960s, at a time when high-quality imaging was not available to publishers. For this reason, we include as a supplement to this article a set of photographs of all Anaxyelinae type specimens from Karatau, taken using modern imaging equipment (see Appendix A).

### 4.5. First Appearance: A Syntexine in Karatau

Over the 100 years between the first fossil insect discoveries in Karatau and our own fieldwork, a total of 25 anaxyelid specimens have been recovered. Most of these have served as type material for 21 species. Remarkably, all previously collected and described specimens belonged exclusively to the subfamily Anaxyelinae. Even more surprising is that the only Anaxyelidae specimen recovered during our recent field campaign turned out to represent a different subfamily altogether.

The specimen, discovered at the Galkino locality, is rather poorly preserved. The pigmentation is almost entirely lost, so the structures can only be discerned under oblique light. This limitation accounts for the low quality of the resulting photographs. Much more information can be extracted from the original fossil, but this requires constant adjustment of the lighting angle. Our reconstruction is based on a combination of images of the part and counterpart, continuously cross-checked against the original specimen under changing illumination.

Despite its mediocre preservation, the specimen represents the first record of Syntexinae in Karatau and the second occurrence of this subfamily in the Jurassic. Given the significant taxonomic implications, we consider the description of a new genus justified, even based on incomplete material.

Rosse-Guillevic et al. [28] proposed five diagnostic characters for distinguishing Syntexinae from Anaxyelinae. According to this thorough character analysis, Syntexinae can be identified based on the following traits:
The Rs+M fork is positioned well beyond the middle of cell 1mcu, or even distal to crossvein 1m-cu (vs. before or near the middle of cell 1mcu). By this character, *Karasyntexis* is unambiguously assignable to Syntexinae.Cell 2r is short, with a length-to-width ratio of less than 1.5 (vs. greater than 1.5). In *Karasyntexis*, this ratio is exactly 1.5.Cell 2r narrows apically (vs. widens). In *Karasyntexis*, the width at the base and apex is equal.Vein 2r-rs joins the pterostigma near its midlength (vs. near the distal quarter). In *Karasyntexis*, it joins the pterostigma near the distal 2/5.Cell 3r is pointed apically (vs. rounded). In the holotype of *K. martynovi*, the exact junction of Rs with the wing margin is not preserved, but the apex appears to have been sharply pointed.

Among these characters, only the first consistently applies across the known diversity of Anaxyelidae; the others are more variable and frequently show exceptions. We find that *Karasyntexis* gen. nov. meets the primary diagnostic criterion for Syntexinae (the distal Rs+M fork) but lies on the boundary between subfamilies for the remaining characters.

The presence of two fully developed r-rs crossveins clearly distinguishes the new genus from *Eosyntexis*, *Sclerosyntexis*, *Orthosyntexis*, *Paraxiphydria*, *Curvitexis*, and *Curiosyntexis* (the latter possessing a rudimentary 1r-rs). Based on the more basal Rs+M branching (proximal to 1m-cu), the new species differs from *Parasyntexis*, *Deresyntexis*, *Hanguksyntexis*, and all of the aforementioned genera with underdeveloped 1r-rs (except *Eosyntexis*). It differs from extant *Syntexis* in its shortened 1-M vein, the shape of cell 2r, and the closed cell r in the hind wing. From *Dolichosyntexis*, it is distinguished by the same shortened 1-M, as well as differences in the shape of the pterostigma and cells 2r and 3r.

Among all syntexine genera, *Karasyntexis* most closely resembles two: *Daosyntexis* from Daohugou (Jurassic of China) and *Hemisyntexis* from Yixian (Cretaceous of China). It differs from *Daosyntexis* by its longer 5-M vein and the absence of Sc in the forewing, and from *Hemisyntexis* by the pterostigmal desclerotization pattern. In *Karasyntexis*, only a small area in the distal half of the pterostigma is desclerotized—a pattern typical of symphytans—while in *Hemisyntexis*, the base of the pterostigma is desclerotized. In ovipositor length, the new genus is intermediate between *Daosyntexis* and *Hemisyntexis*.

## 5. Conclusions

The anaxyelid specimen discovered during our latest expedition to Karatau represents a new genus within the subfamily Syntexinae. *Karasyntexis martynovi* gen. et sp. nov. is the first syntexine known from Karatau and the second representative of this subfamily from the Jurassic. The new genus shows the greatest similarity to *Daosyntexis* and *Hemisyntexis*.

To date, 22 species of Anaxyelidae have been described from Karatau, based on 26 specimens. This high species-to-specimen ratio suggests that the anaxyelid diversity in Karatau shales is far from fully documented and that future excavations are highly likely to yield many additional new species.

The Jurassic Lake of Karatau has been under scientific scrutiny for a century. Over this time, it has produced vast collections amounting to tens of thousands of specimens, and the number of described species exceeds 800. Yet, the site still holds many secrets, and new discoveries will continue to offer valuable material for generations of future researchers.

## Figures and Tables

**Figure 1 insects-16-00943-f001:**
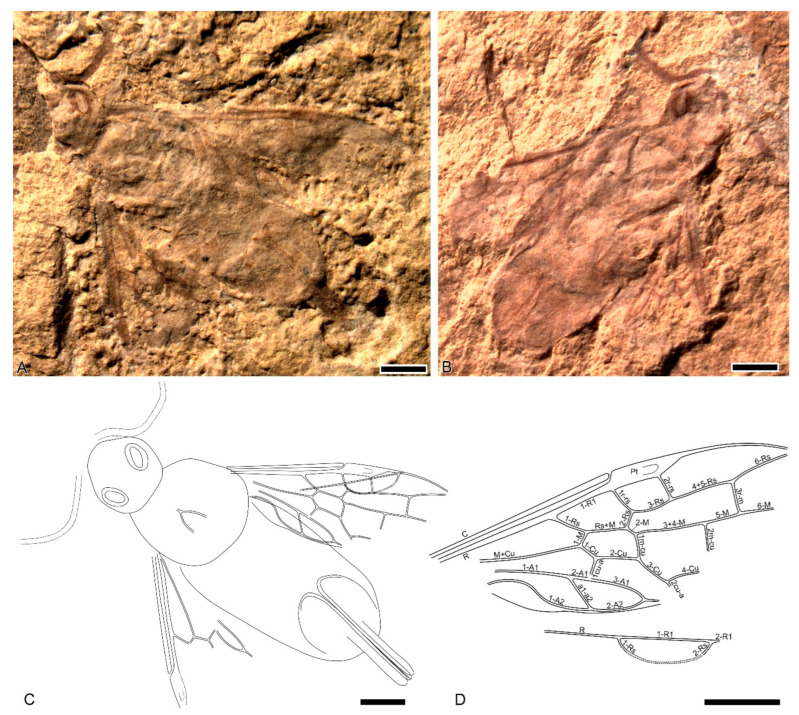
*Karasyntexis martynovi* Kopylov et Rasnitsyn, gen. et sp. nov. Holotype IZRK 1004/213A. (**A**) Part. (**B**) Counterpart. (**C**) Habitus drawing. (**D**) Separated wings (showing names of the veins used in this paper). Scale bars = 1 mm.

**Figure 2 insects-16-00943-f002:**
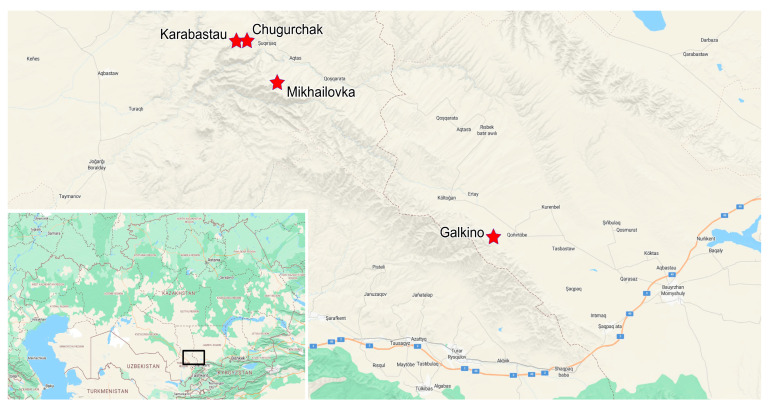
Map showing the location of key sections of the “Jurassic Lake of Karatau”. Basemap: Bing Maps.

**Table 1 insects-16-00943-t001:** List of anaxyelid species from the Jurassic shales of Karatau.

**Anaxyelinae**		
*Anasyntexis strophandra*	Rasnitsyn, 1968	Mikhailovka
*Anaxyela destructa*	Rasnitsyn, 1969	Mikhailovka
*Anaxyela gracilis*	Martynov, 1925	Galkino
*Anaxyela nana*	Rasnitsyn, 1968	Mikhailovka
*Anaxyela parvula*	Rasnitsyn, 1963	Galkino
*Brachysyntexis brachyura*	Rasnitsyn, 1968	Mikhailovka
*Brachysyntexis micrura*	Rasnitsyn, 1969	Mikhailovka
*Brachysyntexis nova*	Rasnitsyn, 1969	Mikhailovka
*Brachysyntexis tenebrosa*	Kopylov, 2018	Mikhailovka
*Brachysyntexis tigris*	Kopylov, 2018	Mikhailovka
*Kulbastavia grandis*	Kopylov, 2018	Mikhailovka
*Kulbastavia macrura*	Rasnitsyn, 1963	Galkino
*Sphenosyntexis antonovi*	Rasnitsyn, 1963	Galkino
*Sphenosyntexis pallicornis*	Rasnitsyn, 1969	Mikhailovka
*Syntexyela asiatica*	Rasnitsyn, 1968	Mikhailovka
*Syntexyela gracilicornis*	Rasnitsyn, 1968	Mikhailovka
*Syntexyela inversa*	Rasnitsyn, 1968	Mikhailovka
*Syntexyela media*	Rasnitsyn, 1963	Galkino
*Urosyntexis depressa*	Rasnitsyn, 1969	Mikhailovka
*Urosyntexis drepanura*	Rasnitsyn, 1968	Mikhailovka
*Urosyntexis magna*	Rasnitsyn, 1968	Mikhailovka
**Syntexinae**		
*Karasyntexis martynovi*	gen. et sp. nov.	Galkino

## Data Availability

Additional data supporting this study, including the original photographs of the newly described species and all type specimens of Anaxyelidae from Karatau previously described in the literature, are openly available on Zenodo at: https://doi.org/10.5281/zenodo.16977336.

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
