# Peer review of "Anaxyelidae of Karatau: 100 Years Afterâ€"

_insects, 2025, doi:10.3390/insects16090943_

Round 1
Reviewer 1 Report
Comments and Suggestions for Authors
The paper “Anaxyelidae of Karatau: 100 Years Later” is a good contribution to the study of the small but interesting family of Anaxyelidae wood wasps that have just one actual species but more extant species described from Jurassic and especially from Early Cretaceous. The subject is maybe a little narrow and from here maybe the not so extended area of the paper with just 33 reference and from them 28 are mentioned in introduction.
Introduction is well made. At line 37 appear the mention of “(Seward, 1907; Nigmatova et al., 2017)” that are not at the references. The formulation “Mikhailovka locality had not yet been discovered” (line 51) is a little confusing because there are many human Mikhailovka localities.
Material and Methods are well presented. At the line 125 is mentioned “Rasnitsyn (1969)”. Must be clarified is on references, possible to be number 10.
The Results are very short, just one page, including the descriptions of a new genus and a new species. The description of the new genus and of the new species are too laconic, too short. I suppose the reason of this is, unfortunately, the poor quality of the fossil remains. In the general aspect of the paper, the shortness of the Results do not look so good, especially if we compare with the almost six pages on the Discussions.
The Discussions (Discussion in the paper) are well made but with an extended part on ,,Current state of key sections of the "Jurassic Lake of Karatau" Lagerstätte” that cover almost more than half of the Discussions. I suggest putting in the title of the paper this, by modifying in a way similar to: Anaxyelidae of Karatau: 100 Years Later and some considerations on “Jurassic Lake of Karatau”.
The supplementary material (Appendix 1) is very big, 624 MB, with a lot of pictures. Many pictures (Anasyntexis strophandra Hol PIN 2066-3369 01.jpg, Anasyntexis strophandra Hol PIN 2066-3369 02.jpg, Anaxyela destructa Hol PIN 2384-1314 01.jpg, Anaxyela gracilis Hol PIN 2452-481 01.jpg, Anaxyela gracilis Hol PIN 2452-481 02.jpg, Anaxyela gracilis Hol PIN 2452-481 03.jpg, Anaxyela gracilis Hol PIN 2452-481 04.jpg, Anaxyela gracilis Spec NA 01.jpg, Anaxyela nana Hol PIN 2554-1300 01.jpg, Anaxyela nana Hol PIN 2554-1300 02.jpg, Anaxyela parvula Hol PIN 2452-583 01.jpg, Anaxyela parvula Hol PIN 2452-583 02.jpg, Brachysyntexis brachyura Hol PIN 2066-3340 01.jpg, Brachysyntexis brachyura Hol PIN 2066-3340 02.jpg, Brachysyntexis micrura Hol PIN 2239-2486 01.jpg, Brachysyntexis nova Hol PIN 2784-1159 01.jpg, Kulbastavia macrura Hol PIN 1789-14 01.jpg, Kulbastavia macrura Hol PIN 1789-14 02.jpg, Sphenosynexis antonovi Hol PIN 2452-591 01.jpg, Sphenosynexis antonovi Hol PIN 2452-591 02.jpg, Sphenosyntexis pallicornis Hol PIN 2784-1162 01.jpg etc.) are copies from some Russian publications. The editor must verify if can be published in this form, if they have the correct copyright, the permission of the publications from they are taken.
Some pictures from the supplementary material are taken from Russian publications with the explications, sometimes even extended explications, in Russian language (Anasyntexis strophandra Hol PIN 2066-3369 01.jpg, Anaxyela gracilis Hol PIN 2452-481 03.jpg, Anaxyela nana Hol PIN 2554-1300 01.jpg etc.). This also can not remain in this form.
Some pictures from the supplementary material (Anasyntexis strophandra Hol PIN 2066-3369 05.jpg, Anasyntexis strophandra Hol PIN 2066-3369 06.jpg, Brachysyntexis micrura Hol PIN 2239-2486 04.jpg, Brachysyntexis micrura Hol PIN 2239-2486 05.jpg, Brachysyntexis tenebrosa Hol PIN 2066-3344 04.jpg, Brachysyntexis tenebrosa Hol PIN 2066-3344 05.jpg, Kulbastavia grandis Hol PIN 2997-5000 05.jpg, Kulbastavia grandis Hol PIN 2997-5000 06.jpg, Kulbastavia grandis Hol PIN 2997-5000 07.jpg, Kulbastavia grandis Hol PIN 2997-5000 08.jpg etc.) are not very scientific made, we can see in pictures some small round objects around the fossil samples.
Some pictures from the supplementary material are put twice, like Kulbastavia grandis Hol PIN 2997-5000 07.jpg and Kulbastavia grandis Hol PIN 2997-5000 08.jpg. I recommend a carefully check of all the pictures.
Some pictures from the supplementary material, looks to be the ones with TIF extension (Brachysyntexis tenebrosa Hol PIN 2066-3344 01.tif, Brachysyntexis tigris Hol PIN 2997-648 01.tif, Kulbastavia grandis Hol PIN 2997-5000 01.tif etc) looks to be original drawings. This must be mentions and specify what is in the pictures.
The pictures with Karasyntexis martynov must be separate from the rest of the pictures. All these pictures are direct in connexion with the paper. I recommend for the HTML document included in the folder of Karasyntexis martynov to use a more common picture format like JPEG or TIF etc.
The supplementary material can not remain in this form. All the material must be organised in one document, preferable a PDF in the final form, where must be arranged all the ORIGINAL pictures in a systematic way and specify what is in every pictures. The same recommendation for the probably original drawings from the supplementary material (probably the pictures with TIF extension).
Some other comments are on the attached manuscript.
The paper can be published with some minor revision.

Author Response
We are deeply grateful to the reviewers for their work and for the constructive criticism of our manuscript. We have carefully considered all the comments and provide our responses below.
Review 1.
Group of comments 1: References issues.
At line 37 appear the mention of “(Seward, 1907; Nigmatova et al., 2017)” that are not at the references.
At the line 125 is mentioned “Rasnitsyn (1969)”. Must be clarified is on references, possible to be number 10
Response 1. Our deepest gratitude to the reviewer for identifying the errors in the references. The comment is absolutely valid, and all the shortcomings have been corrected.
Comment 2. The formulation “Mikhailovka locality had not yet been discovered” (line 51) is a little confusing because there are many human Mikhailovka localities.
Response 2. Yes, you are right, such toponyms are indeed common across the territory of the former USSR, and this can lead to confusion. Typically, when these localities are mentioned only occasionally, expanded names such as Karatau-Mikhailovka, Karatau-Galkino, etc. are used. However, in our article these names occur very frequently, so we chose not to use the expanded forms throughout the text and instead included a clarifying comment in the Introduction: Toponyms such as Mikhailovka, Galkino, and Karabastau are common across the territory of the former USSR and in some cases even refer to other fossil localities; in this work, these names are used exclusively to designate sections within the Karabastau Formation.
Comment 3. The Results are very short, just one page, including the descriptions of a new genus and a new species. The description of the new genus and of the new species are too laconic, too short. I suppose the reason of this is, unfortunately, the poor quality of the fossil remains. In the general aspect of the paper, the shortness of the Results do not look so good, especially if we compare with the almost six pages on the Discussions.
Response 3. The brevity of the description is indeed partly due to the mediocre preservation of the specimen. Moreover, in paleoentomological papers we are accustomed to describing several species at once, and quite often paleoentomological articles turn into one large "Systematic paleontology" section. However, in this case there is only a single specimen, so there is simply not much that can be written here. We placed the extensive comments on the taxonomic position of the new species in the Discussion, as this seemed more consistent. It is also worth noting that many authors use hundreds of words in descriptions to express in very convoluted formulations what can be seen at a glance in the figure. To us, this appears to be a flawed practice, as it obscures the key characters behind a mass of unnecessary wording.
With your permission, we would prefer to leave this section unchanged. Moreover, the new finding is only a small part of our largely review-based paper.
Comment 4. The Discussions (Discussion in the paper) are well made but with an extended part on ,,Current state of key sections of the "Jurassic Lake of Karatau" Lagerstätte” that cover almost more than half of the Discussions. I suggest putting in the title of the paper this, by modifying in a way similar to: Anaxyelidae of Karatau: 100 Years Later and some considerations on “Jurassic Lake of Karatau”.
Response 4. As you have rightly noted, we aim for laconic wording. We would prefer to keep the title short and striking.
Comment 5. The supplementary material (Appendix 1) is very big, 624 MB, with a lot of pictures. Many pictures (Anasyntexis strophandra Hol PIN 2066-3369 01.jpg, Anasyntexis strophandra Hol PIN 2066-3369 02.jpg, Anaxyela destructa Hol PIN 2384-1314 01.jpg, Anaxyela gracilis Hol PIN 2452-481 01.jpg, Anaxyela gracilis Hol PIN 2452-481 02.jpg, Anaxyela gracilis Hol PIN 2452-481 03.jpg, Anaxyela gracilis Hol PIN 2452-481 04.jpg, Anaxyela gracilis Spec NA 01.jpg, Anaxyela nana Hol PIN 2554-1300 01.jpg, Anaxyela nana Hol PIN 2554-1300 02.jpg, Anaxyela parvula Hol PIN 2452-583 01.jpg, Anaxyela parvula Hol PIN 2452-583 02.jpg, Brachysyntexis brachyura Hol PIN 2066-3340 01.jpg, Brachysyntexis brachyura Hol PIN 2066-3340 02.jpg, Brachysyntexis micrura Hol PIN 2239-2486 01.jpg, Brachysyntexis nova Hol PIN 2784-1159 01.jpg, Kulbastavia macrura Hol PIN 1789-14 01.jpg, Kulbastavia macrura Hol PIN 1789-14 02.jpg, Sphenosynexis antonovi Hol PIN 2452-591 01.jpg, Sphenosynexis antonovi Hol PIN 2452-591 02.jpg, Sphenosyntexis pallicornis Hol PIN 2784-1162 01.jpg etc.) are copies from some Russian publications. The editor must verify if can be published in this form, if they have the correct copyright, the permission of the publications from they are taken.
Response 5. You are absolutely right, in the appendix we provided, along with the original photographs, we also include drawings of the corresponding species from the original publications. This image archive is a very important working tool for us, and we would like to keep it as complete as possible for the readers. Not being specialists in copyright law, we would kindly ask the editor to provide a conclusion on this matter. Almost all of the drawings included in the archive were previously published in the works of A.P. Rasnitsyn and D.S. Kopylov, the authors of this article. Some belong to A.V. Martynov and were published 100 years ago – I suppose in such deep antiquity, the copyright holders can no longer be traced. If the editor deems it necessary, we will, of course, remove these drawings from the appendix. But, in any case, a reference to the sources is indeed necessary in the description of the appendix, and it has been added: The archive also contains drawings of specimens previously published (screenshots from publications) [7,12,13,14,20], several original drawings, and a source SVG vector drawing of K. martynovi.
Update 250828: After discussing the copyright issue with the Insects editorial team, we decided to remove the screenshots of figures from previously published papers from the archive. Essentially, this does not affect the main purpose of the archive – to provide previously unpublished photographs of type material from Karatau. The figures remain available to all interested researchers in the respective publications. We are leaving only the original material in the archive.
Comment 6. Some pictures from the supplementary material are taken from Russian publications with the explications, sometimes even extended explications, in Russian language (Anasyntexis strophandra Hol PIN 2066-3369 01.jpg, Anaxyela gracilis Hol PIN 2452-481 03.jpg, Anaxyela nana Hol PIN 2554-1300 01.jpg etc.). This also can not remain in this form.
Response 6. That’s correct. The drawings included in the appendix were taken directly from the original descriptive publications. Wherever possible, we retained the original figure captions, as they provide useful additional information. And yes, in many cases these were publications in Russian. The idea behind the archive is to present raw images, source data for further research. The polished, retouched illustrations are provided in the main figures. In the appendix, we aimed to include the maximum amount of original data with minimal processing.
Update 250828: These pictures are removed (vide supra).
Comment 7. Some pictures from the supplementary material (Anasyntexis strophandra Hol PIN 2066-3369 05.jpg, Anasyntexis strophandra Hol PIN 2066-3369 06.jpg, Brachysyntexis micrura Hol PIN 2239-2486 04.jpg, Brachysyntexis micrura Hol PIN 2239-2486 05.jpg, Brachysyntexis tenebrosa Hol PIN 2066-3344 04.jpg, Brachysyntexis tenebrosa Hol PIN 2066-3344 05.jpg, Kulbastavia grandis Hol PIN 2997-5000 05.jpg, Kulbastavia grandis Hol PIN 2997-5000 06.jpg, Kulbastavia grandis Hol PIN 2997-5000 07.jpg, Kulbastavia grandis Hol PIN 2997-5000 08.jpg etc.) are not very scientific made, we can see in pictures some small round objects around the fossil samples.
Response 7. Small round objects are shotgun pellets. This is a very convenient method when it is necessary to level a stone with an uneven back surface under the lens. I believe that the background that happened to enter the frame is unlikely to reduce the scientific value of the images. And once again: what we provide here are raw images.
Comment 8. Some pictures from the supplementary material are put twice, like Kulbastavia grandis Hol PIN 2997-5000 07.jpg and Kulbastavia grandis Hol PIN 2997-5000 08.jpg. I recommend a carefully check of all the pictures.
Response 8. No, these are two different photographs. Taken from the same angle, but with different lighting angles.
Comment 9. Some pictures from the supplementary material, looks to be the ones with TIF extension (Brachysyntexis tenebrosa Hol PIN 2066-3344 01.tif, Brachysyntexis tigris Hol PIN 2997-648 01.tif, Kulbastavia grandis Hol PIN 2997-5000 01.tif etc) looks to be original drawings. This must be mentions and specify what is in the pictures.
Response 9. Yes, that’s correct. Here we present the original author’s versions of the drawings published in Kopylov, 2018. In the published version of the article, they were provided in a somewhat modified layout and with significantly reduced resolution. Therefore, in this case, for the archive I used not the screenshots from the paper, but the original images.
Update 250828: These pictures are removed (vide supra).
Comment 10. The pictures with Karasyntexis martynov must be separate from the rest of the pictures. All these pictures are direct in connexion with the paper. I recommend for the HTML document included in the folder of Karasyntexis martynov to use a more common picture format like JPEG or TIF etc.
Response 10. I don’t quite understand in what sense the photographs of K. martynovi should be separated from those of other species. In the archive, all photographs are organized into folders, with each folder corresponding to a particular species. As for the HTML file in the folder – I’m not sure what is meant here. There is no such file in the folder. Perhaps the reviewer is referring to the SVG file (if you don’t have Inkscape installed, the system will most likely attempt to open it via a browser, though it is unlikely to succeed). I added a note about the SVG file in the archive description.
Just in case you haven’t worked with this format, let me clarify: SVG is a vector graphics format, a multilayered document that can contain photographs, drawings, measurements, and so on. If you open it in a browser, you’ll see a mess of these layers that looks like a picture. However, this format cannot be reduced to a simple JPG.
Comment 11. The supplementary material can not remain in this form. All the material must be organised in one document, preferable a PDF in the final form, where must be arranged all the ORIGINAL pictures in a systematic way and specify what is in every pictures. The same recommendation for the probably original drawings from the supplementary material (probably the pictures with TIF extension).
Response 11. I would like to clarify that the archive provided is not a set of illustrations created specifically for this project. Rather, it is part of a personal archive on fossil Anaxyelidae that has been compiled over more than 10 years. In fact, the author’s version of the archive contains all known anaxyelid species, including several that are still unpublished, but sharing it in full would be completely unthinkable due to copyright restrictions. Therefore, we selected only the Karatau species, where all illustrations belong either to us (Kopylov, Rasnitsyn) or to long-departed researchers (Martynov). This is a personal working tool, parts of which have been added gradually, photographed with different equipment, under different conditions, and for different purposes. This explains the heterogeneity of the archive: for some specimens there are more photographs, for others fewer; different formats and methods were used at different times. And this is precisely raw data – original photographs in their full quality, in the quantity and form in which we were able to collect them up to the present day. Some of these images have been published in earlier articles, but the majority have remained unseen by readers, stored in the authors’ personal archives. The idea behind this appendix is to give readers access to the very source data the authors themselves work with, in all their original quality and chaos. Converting them into PDF or otherwise “polishing” them would mean sacrificing their original quality and the very concept of publishing source data.
In any case, I see that there are many comments regarding the archive. It is clear that the reviewer supports the idea of publishing the supplement, but has numerous concerns specifically about its formatting. Given all of the above, I would suggest leaving the decision on whether adjustments to the archive are necessary to the editor’s discretion.
Comment 12. Some other comments are on the attached manuscript.
Response 12. Thank you for the careful and thoughtful reading of our text. All the noted comments have been corrected.
Reviewer 2 Report
Comments and Suggestions for Authors
Dear Authors,
Dear Editors,
I am grateful for the opportunity to review this manuscript. The study fills a significant gap in our knowledge of Jurassic insect diversity by documenting the first occurrence of the subfamily Syntexinae in Karatu-Lagerstätte. This site is one of the richest deposits of Jurassic insect fossils in the world, but until now there have been no records of Syntexinae. This discovery not only adds to our knowledge of the diversity of the Anaxyelidae family, and has broader implications for understanding the evolutionary history and biogeography of sircoid wasps. Given that Karatu has been studied for over a century, the discovery of a new genus during recent fieldwork highlights the potential for future discoveries and the importance of continuing palaeontological research at "classic" fossil sites.
This manuscript may be published in the journal Insects after minor revisions.
Below are some comments/recommendations that I hope will help improve the manuscript.
A few comments on inaccuracies in the English text and typos.
Please note the hyphens, which appear to be line break artifacts and should be corrected for final publication. These comments are primarily for the editorial board:
Page 1, line 12 – ‘Ju- rassic’ → delete hyphen → Jurassic.
Page 1, line 15 – ‘prominent’ → delete hyphen → prominent.
Page 1, line 21 – “historically” → delete hyphen → historically.
Page 2, line 40 – ‘discoveries’ → discoveries.
Page 2, line 41 – ‘paleontolog- ical’ → paleontological.
Page 2, line 47 – ‘outstand- ing’ → outstanding.
Page 2, line 54 – ‘synon- ymized’ → synonymized.
Page 2, line 63 – ‘deposits’ → deposits.
Page 3, line 88 – ‘Syn- texinae’ → Syntexinae.
Page 4, line 114 – ‘clean- ing’ → cleaning.
Page 4, line 117 – ‘sur- face’ → surface.
Page 5, line 145 – the sentence ‘Diagnosis:’ is too concise; for clarity, it is recommended to split it into two sentences.
Page 9, line 263 – ‘gra- cilis’ → gracilis.
Please use authors' initials without spaces (A.P., not A. P.) and ensure that journal titles are italicised and written in full or abbreviated in accordance with official abbreviations. Currently, I see both variants in the list.
As far as I know, the titles of books and collections are italicised in accordance with MDPI rules.
In this review, I have focused only on the formal aspects of the manuscript, as from a scientific point of view, this work is completely ready for publication and does not require any changes.
Congratulations to the authors!
Best regards
Author Response
We are deeply grateful to the reviewers for their work and for the constructive criticism of our manuscript. We have carefully considered all the comments and provide our responses below.
Review 2.
Comment 1. Please note the hyphens, which appear to be line break artifacts and should be corrected for final publication.
Response 1. This is very strange. I checked both versions I submitted: the docx and the pdf. Neither of them contains any hyphenation or the typographical errors with dashes mentioned by the reviewer. Most likely, we are dealing with some kind of format incompatibility. From my editorial experience, I know that such glitches sometimes occur when transferring files between different text editors, especially in cross-platform transfers. In any case, thank you for pointing this out, and we hope that everything will display correctly on the editor’s side.
Comment 1.1. Page 5, line 145 – the sentence ‘Diagnosis:’ is too concise; for clarity, it is recommended to split it into two sentences.
Response 1.1. You mean the long sentence with the forewing characters? Usually, when writing diagnoses, different body parts are separated by periods, while characters referring to the same body part are separated by semicolons. Thus, “Forewing with…” is followed by everything pertaining to the forewing. Logically, there is no place to insert a period within that section.
Comment 2. Please use authors' initials without spaces (A.P., not A. P.) and ensure that journal titles are italicised and written in full or abbreviated in accordance with official abbreviations. Currently, I see both variants in the list.
Response 2. It seems to be another software compatibility issue. I searched my entire file using the wildcard “[A-Z]. [A-Z].” and did not find a single instance. As for italics in journal titles – indeed, I corrected them in several places.
Comment 3. As far as I know, the titles of books and collections are italicised in accordance with MDPI rules.
Response 3. Yes, that’s correct. It seems that in my list all book titles are given in italics, except for the monograph on Khasurty, but it is formally an article rather than a book.
Reviewer 3 Report
Comments and Suggestions for Authors
Anaxyelidae of Karatau: 100 Years Later D.S. Kopylov, and A.P. Rasnitsyn
Line 93 In this paper, we describe a new genus and species (and a new subfamily for Karatau)
Suggestion: In this paper, we describe a new genus and species (and a new subfamily record for Karatau).
105 A detailed characterization of the localities is provided in the Discussion.
Comment. Normally the description of sites like this could go in the methods section, but the final decision is in the hands of the authors and/or editor.
108 – 124 Extraction methods.
130 Figure 1. Karasyntexis martynovi Kopylov et Rasnitsyn, sp. Nov
Must be: Figure 1. Karasyntexis martynovi Kopylov et Rasnitsyn, gen. nov. et sp. Nov.?
143 As stated in the summary (line 18), it is a new species. The name of the species is placed here (143) as already described in 2020. I thought there was an error (see line 159). Should be: Karasyntexis martynovi Kopylov et Rasnitsyn, sp. nov. I do not have access to the publication by Kopylov et al. 2020 to verify this species name. Please verify.
145 and ss. I don't know the wing nomenclature of basal hymenoptera, let alone fossils, but we assume it is correct based on Alex's experience.
Comment. Since I am not a paleontologist I cannot comment on this section, but with Alex's great experience I think everything is fine.
307 and ss. As the authors point out, the fossil has been poorly represented, making it difficult to make decisions about its taxonomic range. I suppose this name should remain available until new material can be found, hopefully in better condition.
341 change: … that Karasyntexis meets the primary diagnostic …
By: … that Karasyntexis gen. nov. meets the primary diagnostic …
General comment
Hymenoptera fossil finds from the Jurassic are relatively poor, so this description of a new symphyte is welcome, even if the fossil is not well preserved. Furthermore, the Syntexinae subfamily has only one known Jurassic record, and none for the locality, which increases the importance of the find.
On the other hand, I applaud the efforts of the authors to search for, preserve and describe (when possible) new taxa, especially due to the difficulties of obtaining collection permits and the presence of illegal fossil hunters.
The high-quality photos of holotypes of Anaxyelidae from Karatau are an important addition to this manuscript, very useful for the study of the evolution and diversity of Mesozoic Hymenoptera.
Author Response
We are deeply grateful to the reviewers for their work and for the constructive criticism of our manuscript. We have carefully considered all the comments and provide our responses below.
Reviewer 3.
Comment 1. Line 93 In this paper, we describe a new genus and species (and a new subfamily for Karatau). Suggestion: In this paper, we describe a new genus and species (and a new subfamily record for Karatau).
Response 1. Thank you, that’s a good suggestion. I accept it.
Comment 2. 105 A detailed characterization of the localities is provided in the Discussion. Comment. Normally the description of sites like this could go in the methods section, but the final decision is in the hands of the authors and/or editor.
Response 2. In this case, the locality is not just a source of material but also a subject of discussion. Therefore, it seemed logical to me to move my considerations about Karatau into the corresponding section.
Comment 3. 108 – 124 Extraction methods.
Response 3. Accepted.
Comment 4. 130 Figure 1. Karasyntexis martynovi Kopylov et Rasnitsyn, sp. Nov Must be: Figure 1. Karasyntexis martynovi Kopylov et Rasnitsyn, gen. nov. et sp. Nov.?
Response 4. Thank you! Corrected.
Comment 5. 143 As stated in the summary (line 18), it is a new species. The name of the species is placed here (143) as already described in 2020. I thought there was an error (see line 159). Should be: Karasyntexis martynovi Kopylov et Rasnitsyn, sp. nov. I do not have access to the publication by Kopylov et al. 2020 to verify this species name. Please verify.
Response 5. Thank you, it is very fortunate that you noticed this! Our original plan was to describe the species within the genus Daosyntexis, but we later realized that the differences were too significant, and we had to revise an almost completed manuscript. Now, remnants of that rejected initial idea are surfacing. Corrected.
Comment 6. 145 and ss. I don't know the wing nomenclature of basal hymenoptera, let alone fossils, but we assume it is correct based on Alex's experience. Comment. Since I am not a paleontologist I cannot comment on this section, but with Alex's great experience I think everything is fine.
Response 6. We also very much hope that we have not made any mistakes :)
Comment 7. 307 and ss. As the authors point out, the fossil has been poorly represented, making it difficult to make decisions about its taxonomic range. I suppose this name should remain available until new material can be found, hopefully in better condition.
Response 7. Not quite. The specimen is indeed not in the best state of preservation. However, the key characters used to distinguish the subfamilies are preserved, and for the most part they lie on the borderline between the two subfamilies, with the exception of the first character (the Rs+M fork), which is the most reliable and by which the specimen is unambiguously assigned to Syntexinae.
Comment 8. 341 change: … that Karasyntexis meets the primary diagnostic … By: … that Karasyntexis gen. nov. meets the primary diagnostic …
Response 8. Done.
Round 2
Reviewer 1 Report
Comments and Suggestions for Authors
The authors corrected all the problems. They also resolved the problem with the copyright of some pictures taken from other publications by remove them.
Remain the problem with the archive. The authors argue that they want to put the RAW pictures in the archive and even one SVG vectorial graphic. So, the archive has more than 600 MB. I understand that they want the maximum quality of the pictures and don’t accept to use a more convenient way like a PDF document. Imagine that the all the authors want the maximum quality of the pictures and want to have such big archives of hundreds of megabits or even terabytes attached to every articles. This is not a common practice seen in articles. As the authors suggest I will let this problem to the editors.
The paper can now be published.
Author Response
Comment 1. Remain the problem with the archive. The authors argue that they want to put the RAW pictures in the archive and even one SVG vectorial graphic. So, the archive has more than 600 MB. I understand that they want the maximum quality of the pictures and don’t accept to use a more convenient way like a PDF document. Imagine that the all the authors want the maximum quality of the pictures and want to have such big archives of hundreds of megabits or even terabytes attached to every articles. This is not a common practice seen in articles. As the authors suggest I will let this problem to the editors.
Response 1. I'm sure that modern technology allows the transfer of such file sizes within minutes, and it hardly seems justified to sacrifice quality for the sake of bandwidth. The possibility of publishing large online supplements has only become available to researchers in recent years, and many are not yet accustomed to this remarkable opportunity. Personally, I strongly support this practice and hope it will become a standard in our field. As a specialist in the systematics of Mesozoic Hymenoptera, I often find myself in need of precisely such raw data - particularly collections of primary images of described species - which are usually absent from colleagues’ papers. By providing such data myself, I hope to encourage others to share more of the material that is vital to our work. I trust in your understanding. And despite these disagreements, I am sincerely grateful to the Reviewer for their thorough work and constructive criticism.